# Heterologous Expression of the Grapevine *JAZ7* Gene in Arabidopsis Confers Enhanced Resistance to Powdery Mildew but Not to *Botrytis cinerea*

**DOI:** 10.3390/ijms19123889

**Published:** 2018-12-05

**Authors:** Muhammad Hanif, Mati Ur Rahman, Min Gao, Jinhua Yang, Bilal Ahmad, Xiaoxiao Yan, Xiping Wang

**Affiliations:** 1State Key Laboratory of Crop Stress Biology in Arid Areas, College of Horticulture, Northwest A&F University, Yangling 712100, China; mhanif@nwafu.edu.cn (M.H.); mati@nwafu.edu.cn (M.U.R.); gaomin@umbc.edu (M.G.); 13759927085@163.com (J.Y.); bajwa1999@nwafu.edu.cn (B.A.); xiaoxyan@nwafu.edu.cn (X.Y.); 2Key Laboratory of Horticultural Plant Biology and Germplasm Innovation in Northwest China, Ministry of Agriculture, Northwest A&F University, Yangling 712100, China

**Keywords:** *VqJAZ7*, grapevine, *Golovinomyces cichoracearum*, powdery mildew, *Botrytis cinerea*, *Pst* DC3000

## Abstract

Jasmonate ZIM-domain (*JAZ*) family proteins comprise a class of transcriptional repressors that silence jasmonate-inducible genes. Although a considerable amount of research has been carried out on this gene family, there is still very little information available on the role of specific *JAZ* gene members in multiple pathogen resistance, especially in non-model species. In this study, we investigated the potential resistance function of the *VqJAZ7* gene from a disease-resistant wild grapevine, *Vitis quinquangularis* cv. “Shang-24”, through heterologous expression in *Arabidopsis thaliana*. *VqJAZ7*-expressing transgenic Arabidopsis were challenged with three pathogens: the biotrophic fungus *Golovinomyces cichoracearum*, necrotrophic fungus *Botrytis cinerea*, and semi-biotrophic bacteria *Pseudomonas syringae pv. tomato* DC3000. We found that plants expressing *VqJAZ7* showed greatly reduced disease symptoms for *G. cichoracearum*, but not for *B. cinerea* or *P. syringae*. In response to *G cichoracearum* infection, *VqJAZ7*-expressing transgenic lines exhibited markedly higher levels of cell death, superoxide anions (O_2_¯, and H_2_O_2_ accumulation, relative to nontransgenic control plants. Moreover, we also tested the relative expression of defense-related genes to comprehend the possible induced pathways. Taken together, our results suggest that *VqJAZ7* in grapevine participates in molecular pathways of resistance to *G. cichoracearum*, but not to *B. cinerea* or *P. syringae*.

## 1. Introduction

Plants use physical and chemical barriers as the first line of defense against invading pathogens. However, inducible molecular defense strategies are also important for plant survival. These defensive barriers comprise a complex immune system known as innate immunity to defend the plant against pathogen infection. Plasma membrane proteins that are designated pattern-recognition receptors (PRRs) recognize pathogen-associated molecular patterns (PAMPs) and induce a basal resistance response called PAMP-triggered immunity (PTI) [1]. Phytopathogens can suppress PTI using a battery of virulent effector molecules [2,3]. These effector molecules are recognized by a family of polymorphic intracellular receptor complexes called nucleotide-binding/leucine-rich repeat (NLR/NB-LRR) and induce a second phase of the resistance response called effector-triggered immunity (ETI) [4]. ETI is strong, robust, and leads to the hypersensitive response (HR), which is a type of programmed cell death (PCD) [5], which decreases the proliferation of the attacking pathogen.

Key signaling molecules in plant defense include the endogenous hormones salicylic acid (SA) and jasmonic acid (JA) [6]. SA-dependent signaling pathways are deployed in response to a broad range of pathogens [7], particularly the biotrophs [8]. JA, in addition to pathogen infection [9,10,11,12], works as a defense signal to stress [13], wound responses [14,15,16], and insect attacks [17,18]. JA also has several endogenous regulatory roles in diverse plant processes, including stamen development [19,20,21,22,23], root growth [24,25,26], anthocyanin accumulation [27,28], and senescence [29,30,31]. The SA and JA pathways can interact, both positively and antagonistically. In *Arabidopsis thaliana* (Arabidopsis), the *eds4* and *pad4* mutants, which are impaired in the accumulation of SA, showed enhanced JA-dependent gene expression [32], and other evidence indicates that SA signaling can inhibit JA signaling in tomato [33,34]. On the other hand, the treatment of tobacco plants with *Erwinia carotovora* elicitors inhibited the expression of SA-related genes [35], suggesting that JA can negatively influence SA signaling. It is also evident from the literature that SA and JA-related genes can be expressed synergistically. For instance, in a microarray study in which Arabidopsis plants were subjected to different defense-inducing treatments, more than 50 genes were regulated by both SA and JA-mediated signaling [36]. Antagonism between SA and JA-mediated signaling pathways might ensure appropriate responses to certain pathogens, for instance, the ability of SA to promote PCD might be prevented by the plant in the case of necrotrophic pathogens, which use cell death for its own benefit [6].

Jasmonate ZIM-domain (JAZ)-family proteins act as negative regulators of JA signaling through repressive physical interaction with *MYC2* transcription factors [37,38,39] and other proteins such as the corepressor TOPLESS (TPL) [26]. When the plant is exposed to abiotic or biotic stress, the JA content in the cell increases, and *JA-Ile* (Jasmonoyl-L-isoleucine conjugate), the active form of JA, promotes physical interaction between JAZ proteins and coronatine-insensitive 1 (COI1), which targets the proteins to the 26S proteasome. The proteolytic degradation of JAZ proteins by the proteasome thus derepresses JA-mediated transcription and downstream processes [37,38,39,40,41].

Since their initial characterization, a substantial amount of research has been done on the function of JAZ genes in reference plants such as Arabidopsis and rice, but only a few studies have focused on horticultural crops such as grape. Grape is one of the most important fruit crops worldwide and is consumed in many forms, i.e., fresh, dried, juice, vinegar, and wine. Most of the high-yielding, improved grape varieties are interspecific hybrids of *Vitis vinifera*, a species that is highly susceptible to diseases and nematodes. Disease is an important factor limiting yield in grapes, and is commonly addressed by the prodigious use of pesticides. However, pesticides and other chemical inputs increase the cost of production, and can have adverse effects on human health. The use of disease-resistant cultivars is an alternative approach, and in-depth knowledge about defense-related genes and their associated molecular pathways is essential for molecular breeding programs to provide disease-resistance cultivars.

In a previous study, we identified the *JAZ7* gene from the common grapevine, *Vitis vinifera*, as highly induced by various biotic stresses [42]. We identified 11 *JAZ* genes in the grapevine genome and named them *VvJAZ1–VvJAZ11*. We found that these genes showed diverse transcriptional responses to exogenous SA and JA, as well as two additional hormones, ethylene and abscisic acid. Interestingly, the expression of *VvJAZ7* was upregulated in response to all four of these hormones. There is very little information available about the function of *JAZ7* [43]. In this study, we engineered transgenic Arabidopsis plants expressing the *VqJAZ7* gene from the disease-resistant Chinese wild grape *V. quinquangularis* cv. “Shang-24”, and evaluated the plants for resistance to the biotrophic fungus *Golovinomyces cichoracearum* (a causal agent of powdery mildew), necrotrophic fungus *Botrytis cinerea* (gray mold), and semi-biotrophic pathogen *Pseudomonas syringae* pv. tomato DC3000.

## 2. Results

### 2.1. Bioinformatic Information about VqJAZ7 Gene

To gain insight into the potential functional relationships among JAZ proteins, we carried out a phylogenetic analysis of JAZ proteins from grapevine, Arabidopsis, and rice (Figure 1). *VvJAZ7* was grouped with *VvJAZ8* within a well-defined clade also containing *VvJAZ1*, 3, 5, 6, 8, and 11, as well as Arabidopsis *JAZ3*, 4, and 9. No rice JAZ member was found within this clade (Figure 1). Other information regarding multiple alignment of the ORF translation sequences of *JAZ7* genes from *V. quinquangularis* and *V. vinifera*, multiple alignment of ORF translations of *JAZ7* genes from *V. quinquangularis* and *V. vinifera*, and the ORF sequence of the *VqJAZ7* gene and its translation were provided in Appendix A.

### 2.2. Analysis of VqJAZ7 Overexpression Lines

To evaluate a potential role for *VqJAZ7* in pathogen defense, we engineered transgenic Arabidopsis constitutively expressing *VqJAZ7*, and challenged the plants by inoculation with *G. cichoracearum*, the causative agent of powdery mildew disease in grape. Eighteen independent transgenic lines were produced, and three highly resistant transgenic lines (TL1, TL2, and TL3) were selected to generate T3 homozygous lines. The transgenic lines were confirmed by PCR (Appendix A). The expression of *VqJAZ7* was evaluated in these three transgenic lines after *G. cichoracearum* inoculation. We found that, even though *VqJAZ7* was engineered to be expressed constitutively in the transgenic lines, *VqJAZ7* transcript levels increased after infection, reaching a maximum of ~4.6-fold increase at three days post-inoculation (dpi) and then declining (Figure 2).

### 2.3. Arabidopsis VqJAZ7 Transgenic Lines Show Enhanced Resistance to Powdery Mildew

To evaluate the potential for *VqJAZ7* in defense against *G. cichoracearum,* we inoculated plants with the pathogen and evaluated the effects at seven dpi. In *VqJAZ7*-expressing transgenic lines, fewer disease symptoms were observed than in nontransgenic controls (Figure 3A). To explore the mechanisms by which enhanced resistance to *G. cichoracearum* might be imparted, we stained leaves with trypan blue, nitro blue tetrazolium (NBT), and diaminobenzidine (DAB) to assess levels of cell death, superoxide anions (O_2_¯), and H_2_O_2_, respectively. As shown in Figure 3, *VqJAZ7*-expressing transgenic lines exhibited markedly higher levels of cell death, superoxide anions (O_2_¯), and H_2_O_2_ accumulation, relative to nontransgenic control plants.

To help define the molecular pathways by which *VvJAZ7* may contribute powdery mildew (PM) resistance to Arabidopsis, we evaluated the expression levels of a cadre of genes previously shown to be SA-responsive or JA-responsive, in response to PM infection (Figure 4). The SA-dependent disease resistance genes *isochorismate synthase 1* (*AtICS1*) and *pathogenesis-related gene 1* (*AtPR1*), were significantly upregulated after PM infection in transgenic plants (Figure 4A). The SA signal is transduced via NPR1, which interacts with transcription factors participating in SA-mediated pathogenesis related (PR) gene expression [44,45], and *PR1* is a marker gene for systemic acquired resistance (SAR) in Arabidopsis [46]. In our study, the expression level of *AtPR1* was downregulated in the nontransgenic control plants; however, its expression was significantly upregulated, ~threefold at one dpi and sevenfold at later time points in the transgenic lines when compared to control plants. *AtICS1* is involved in the biosynthesis and accumulation of SA [47]. We found that its expression was upregulated both in the transgenic and control plants, but the expression in control plants was still very low relative to transgenic lines. *AtICS1* was upregulated ~4.5-fold, 2.8-fold and 5.7-fold at one, three, and five dpi, respectively, in transgenic lines as compared to control plants (Figure 4A). Furthermore, the transcript levels of *lipoxygenase-3* (*AtLOX3*), which is involved in JA formation [48,49], was downregulated at all time points, with ~5.8-fold down-regulation at 5 dpi. *Plant defensin 1.2* (*AtPDF1.2*), which mediates methyl jasmonates (MeJA) signaling [50], was also downregulated, 8.3-fold at one dpi to 6.1-fold at five dpi, as compared to nontransgenic control plants (Figure 4B).

### 2.4. Heterologous Expression of VqJAZ7 in Arabidopsis Decreased Resistance to B. Cinerea

To assess the effect of the heterologous expression of *VqJAZ7* regarding defense to other pathogens, we challenged transgenic plants with the necrotrophic fungal pathogen, *B. cinerea*. At 72 h post-inoculation (hpi), we noted lesions on all of the tested plants; however, transgenic lines showed more prominent disease lesions than nontransgenic control plants (Figure 5A), with bigger lesion diameters (Figure 5D). In the transgenic lines, the proportion of large lesions was significantly greater than in the nontransgenic controls (Figure 5E). Since necrotrophic pathogens ultimately kill infected cells, we performed the histochemical staining of infected plants. Compared with controls, more cell death and H_2_O_2_ production were evident in leaves from transgenic plants (Figure 5B,C). Moreover, the activities of antioxidant enzymes catalase (CAT), peroxidase (POD), and superoxide dismutase (SOD) were lower in the leaves of the transgenic plants. We noted highest values of CAT, POD, and SOD at 48 hpi, 72 hpi, and 24 hpi, respectively (Figure 5F–H).

To gain insight into the apparent decrease in resistance to *B. cinerea* conferred by *VqJAZ7*, we evaluated the expression of the defense-related genes *AtPR1*, *AtICS1*, *AtPDF1.2,* and *AtLOX3* in response to *B. cinerea* at 0 hpi, 24 hpi, 48 hpi, and 72 hpi (Figure 6) in transgenic and control plants. *AtICS1* was expressed to 9.9-fold and sevenfold higher levels in *VqJAZ7* transgenic lines compared to controls at 48 hpi and 72 hpi (Figure 6A). Likewise, the expression of *AtPR1* was elevated 4.4-fold and 4.1-fold at 48 hpi and 72 hpi, respectively, in the transgenic plants. In contrast, the expression of JA-related genes, *AtPDF1.2* and *AtLOX3*, was significantly lower in transgenic lines as compared to controls (Figure 6B) at all of the time points post-infection. *AtPDF1.2* and *AtLOX3* were both expressed differentially, but this expression was lower than that of the control plants.

### 2.5. Heterologous Expression of VqJAZ7 in Arabidopsis Decreased Resistance to Pst DC3000

To investigate a potential influence of *VqJAZ7* on resistance to *Pst* DC3000, transgenic Arabidopsis lines overexpressing *VqJAZ7* and nontransgenic control plants were inoculated with *Pst* DC3000. All of the plants were examined 72 hpi. Relative to nontransgenic control plants, more disease symptoms were observed in the *VqJAZ7*-expressing plants (Figure 7A). We noted less cell death in the transgenic lines as measured by trypan blue staining (Figure 7B). We also observed lower O_2_¯ accumulation in the transgenic lines, as evaluated by NBT staining (Figure 7C). This might result from decreased cell death in the transgenic lines. We counted the bacterial population per unit of leaf area (Figure 7D) to assess the growth of the pathogen at 72 hpi. As shown in Figure 7, higher numbers of bacteria were found on the leaves of the transgenic lines as compared with controls.

We also evaluated the expression of *AtPR1*, *AtICS1*, *AtPDF1.2* and *AtLOX3* in response to *Pst DC3000*, at 0 hpi, 24 hpi, 48 hpi, and 72 hpi, (Figure 8), in transgenic and control plants. In nontransgenic control plants, *AtICS1* showed strong and transient induction at 48 hpi, with decreased expression at 72 hpi (Figure 8). In all three transgenic lines, *AtICS1* was strongly induced by 24 hpi, with decreased expression at subsequent time points. *AtPR1*, *AtPDF1.2*, and *AtLOX3* were all strongly induced by 24 hpi in nontransgenic control plants (Figure 8). However, these genes were not induced, or induced to substantially decreased levels, in all three transgenic lines.

### 2.6. Expressing VqJAZ7 in Arabidopsis Decreased Elicitor-Dependent Callose Accumulation

The plant cell wall acts a barrier against attacking pathogens. Upon stress stimuli, plants modify and remodel the cell wall to incorporate defense-related polysaccharides, such as the 1,3-β-glucan callose [51]. We applied *Pst* DC3000 bacteria or the defense response elicitors flg22 and lipopolysaccharide (LPS) to the control and transgenic plants and stained them with aniline blue. As shown in Figure 9, the control plants accumulated substantially more callose than the transgenic lines.

## 3. Materials and Methods

### 3.1. Plant and Pathogen Material and Growth Conditions

*Vitis quinquangularis* cv. “Shang-24” plants were maintained in a curated *Vitis* collection at Northwest A&F University, Yangling, Shaanxi, China. Two-year-old plants were used in this study. Wild-type *Arabidopsis thaliana* Col-0 plants were grown at 21 °C and 60% relative humidity in long-day photoperiods (16 h-light and eight h-dark). Arabidopsis *pad4* mutant plants were also grown under these conditions for the culturing of *G. cichoracearum*. *B. cinerea* was isolated from *V. vinifera* cv. Red Globe and propagated on potato glucose agar at 22 °C. The *P. syringae* pv. tomato DC3000 strain (*Pst* DC3000) was stored for long-term use at −80 °C. 

### 3.2. Bioinformatic Analyses

Curated open reading frame (ORF) translations from Arabidopsis, grapevine, and rice were as maintained by the National Center for Biotechnological Information (http://blast.ncbi.nlm.nih.gov/Blast.cgi) and the J. Craig Venture Institute (http://www.tigr.org/). Sequences were aligned and the phylogenetic tree was constructed using the neighbor joining (NJ) method in ClustalX2 (http://www.clustal.org/clustal2/). The tree was designed using an online tool (Interactive Tree Of Life, Version 3; (http://itol.embl.de/) [52,53]. The bootstrap values were calculated with 1000 replicates.

### 3.3. Extraction of RNA from Grapevine Leaves and Semi-Quantitative RT-PCR

Total RNA was extracted from *V. quinquangularis* cv. “Shang-24” according to the protocol of Huang et al., 2016 [54]. First strand cDNA was synthesized from two μg of total RNA with the Primer Script™ RTase according to the supplied protocol (TaKaRa Bio Inc., Dalian, China). cDNA was amplified using 2× Taq PCR Master Mix (Bio Sci Biotech, Hangzhou, China) and oligonucleotide primers designed to specifically amplify the *VqJAZ* or *VvJAZ* transcripts (Appendix A). Grapevine *ACTIN1* was used as a reference gene. Reactions were conducted with the following parameters: 94 °C for eight minutes, followed by 30–40 cycles of 92 °C for 30 s, 58–64 °C for 30 s, 72 °C for 30 s, and a final extension at 72 °C for eight minutes. PCR products were separated on a 1% (*w*/*v*) agarose gel and imaged under UV light to analyze gene expression. All of the reactions were repeated three times.

### 3.4. Plant Transformation

The *VqJAZ7* full-length coding sequence (417 bp) was amplified by PCR using gene-specific primers F1 (5′-GAGGAAGCCCCCAAGAGCAACA-3′) and R1 (5′- TTC TGA AGG AAC CGA TGG AGC G-3′) and 2× Taq PCR Master Mix (Bio Sci Biotech, Hangzhou, China). The resulting product was cloned into the pGEM^®^-T Easy vector (Promega, Madison, WI, USA), and cloned sequence was confirmed as error-free by sequencing. This plasmid was named pGEM-T/*VqJAZ7*. Subsequently, the *VqJAZ7* coding sequence was amplified from the pGEM-T/*VqJAZ7* plasmid using primers F2 and R2 (Appendix A), which introduced *Kpn*I and *Xba*I restriction sites within F2 (5′-CGC TCTAGAATGGAGTTTACCCCCAATC-3′ *Xba*I site underlined) and R2 (5′-GGCGGTACCTTAATGATTGTATGGGGACAT-3′ *Kpn*I site underlined), respectively. After the *VqJAZ7* coding sequence had been excised from pGEM-T/*VqJAZ7* and purified, it was inserted downstream of the CaMV 35S promoter in the plant overexpression vector pCambia2300 (Cambia, Brisbane, QLD, Australia), and this plasmid was introduced into Arabidopsis by the floral dip method [55]. T_1_ seeds were collected and sown on MS medium containing 10 g/L sucrose, eight g/L agar at pH 5.8, supplemented with 75 mg/L kanamycin to screen transgenic lines. A total of 18 independent lines were obtained, and the three lines showing the strongest apparent resistance to PM (TL1, TL2, and TL3) were selected for further study. Presence of the transgene was confirmed with PCR. Homozygous T3 lines were generated for all further experiments.

### 3.5. Pathogen Inoculation Assays

*G. cichoracearum* was maintained on live plants of *phytoalexin deficient 4* (*pad4*), a PM-susceptible Arabidopsis mutant [56]. Transgenic and nontransgenic control plants were inoculated with *G. cichoracearum* by the method of Guo et al. [57]. Three days following inoculation, plants were transferred to an ambient environment of 22 °C and 40% RH. Samples were collected at 0, one day, three days, five days, and seven days post-inoculation (dpi) for gene expression analysis. Visual scoring of disease severity and spore counting were performed as described previously [58].

*B. cinerea* was cultured on potato dextrose agar medium at 25 °C in the dark. Conidial spore suspension was prepared in distilled water as described previously [59]. Leaves from 21 four-week-old transgenic and nontransgenic control plants were placed on wet filter paper set on 1% agarose in a tray. The leaves were inoculated by dropping 10 μL of spore suspension (2 × 10^6^ conidia/mL) onto the adaxial surface, and the trays were then sealed with a transparent plastic film to maintain high humidity. Samples were collected at 0 hpi, 24 hpi, 48 hpi, and 72 hpi, and lesion diameter was determined at 72 hpi. Measurements of lesion diameter and antioxidant enzyme activity were carried out as described by Guo et al. [57].

*Pst* DC3000 was grown according to the method of Tornero and Dangl [60] in a rotating incubator at 280 rpm and 28 °C in King’s B medium containing 50 mg/mL rifampicin. When cultures reached an OD_600_ of 0.8–1.0, they were subjected to centrifugation at 5000× *g* for 10 min, and cells were resuspended in 10 mM of MgCl_2_ to an OD_600_ of 0.02. Four-week-old plants were dipped into the cell suspension containing 0.02% Silwet L-77 for 10 minutes, as described previously [60]. Plants were incubated at 90% relative humidity for 24 h, and then transferred to ambient growth conditions. Disease severity was evaluated at three dpi.

To detect callose accumulation, four-week old plants were subjected to pressure infiltration with flg22, *Pst* DC3000, or LPS with a needleless syringe. MgCl_2_ (10 mM) was used for mock inoculation. Samples were collected 18 h later, and stained with aniline blue as described by Wang et al. [61]. 

### 3.6. Histochemical Detection

Histochemical detection was carried out as according to Yan et al. [59] using nine leaves each from the transgenic lines and nontransgenic control. To visualize cell death, leaves were subjected to staining with trypan blue (TB); O_2_^−^ accumulation was assessed using six mM of nitro blue tetrazolium (NBT); H_2_O_2_ accumulation was visualized with one mg·mL^−1^ diaminobenzidine (DAB, pH 3.8). The TB buffer consisted of 10 mL of phenol, 10 mL of lactic acid, 10 mL of ddH_2_O, 20 mL of ethanol, and 10 mg of trypan blue. Leaves were first boiled in TB solution for three minutes, and then decolorized in 2.5 mg·mL^−1^ of chloral hydrate for 48 h. NBT (6 mM) was dissolved in HEPES buffer, pH 7.5. Samples were soaked in NBT solution for two hours and in DAB solution for eight hours, and then decolorized in 80% ethanol for two hours at 60 °C. Samples were kept at room temperature for 48 h, and finally transferred to 50% glycerol for histochemical studies.

### 3.7. Quantitative Real-Time PCR

Leaf samples were harvested at 0 dpi, one dpi, three dpi, five dpi, and seven dpi for PM; and 0 hpi, 24 hpi, 48 hpi, and 72 hpi for *B. cinerea*; and 0 hpi, 24 hpi, 48 hpi, and 72 hpi for *Pst* DC3000. Three biological replicates were collected from each treatment at each time point and immediately frozen in liquid nitrogen. Total RNA was extracted from the samples using the E.Z.N.A.^®^ Plant RNA Kit (Omega Bio-tech, Norcross, GA, USA), and first-strand cDNA was synthesized using PrimerScript^TM^ RTase (TaKaRa BioInc., Dalian, China). Quantitative real-time PCR was performed in triplicate in a final reaction volume of 20 µL, which contained one µL of a six fold cDNA dilution as template and 2× *Taq* PCR MasterMix (BioSci Biotech, Hangzhou, China). Assays were carried out using a Step OnePlus real-time PCR system (Applied Biosystems, Life Technologies, Singapore) and the following thermal profile: 94 °C for 30 s, followed by 35 cycles of 94 °C for 5 s, and 60 °C for 30 s. The Arabidopsis *ACTIN2* gene was used as an internal reference.

### 3.8. Statistical Analysis

Data analysis was performed using Microsoft Excel and Sigma Plot (10.0, Systat, Inc., San Jose, CA, USA). Significance of differences were assessed by paired *t*-tests through one-way ANOVA using SPSS Statistics 17.0 software (IBM China Company Ltd., Beijing, China).

## 4. Discussion

*JAZ* family proteins comprise a small family of 13 members in Arabidopsis [62], and have been characterized as transcriptional repressors of early response genes in JA signaling. *JAZ* proteins consist of two highly conserved motifs: the Jas motif, which is required for *JAZ*–COI1 interactions and degradation in response to *JA-Ile* [38,63,64,65], and the *TIFY/ZIM* domain, which is required for *JAZ* homodimerization and heterodimerization [63]. The repression of gene expression occurs at low levels of JA, because *MYCs* transcription factors are bound to the *JAZ* proteins with the help of corepressors, such as NINJA or TPL [14,26,66]. After the perception of an external stimulus, the amount of active JAs (*JA-Ile*) increases, and the *JAZ* proteins are bound by the F-box signal receptor COI1, and are thereby tagged for degradation by the 26S proteasome [39,67]. Upon *JAZs*’ degradation, MYC proteins are freed to activate the JA signaling and responses [68]. *JAZ7* is one of the most enigmatic members among the 13 *JAZs* in Arabidopsis [69], because it does not form homodimers or heterodimers, and its interaction is limited to very few transcription factors, e.g., *MYC3* and *MYC4* [70,71]. Moreover, *JAZ7* contains the highly conserved EAR motif [72], suggesting that *JAZ7* may not require NINJA to recruit TPL [73]. Direct interaction has been identified between TPL and *JAZ8*, a close homolog of *JAZ7*, by the Arabidopsis interactome study [74].

Plants are exposed to biotic and abiotic stresses throughout their life, and due to their sessile nature, they have evolved complex defense mechanisms to survive. Plants respond to stresses by a simple way, but it recruits a complex network of antagonistic or synergistic nature between SA and JA for its immune system [75,76].

In this study, we investigated the potential role of grapevine *JAZ7* in defense against *G. cichoracearum*, *B. cinerea*, and *Pst* DC3000, through its heterologous expression in Arabidopsis. As for as the biotrophic fungal disease powdery mildew, fewer disease symptoms were observed in transgenic lines compared with nontransgenic controls at seven dpi (Figure 3A). This result was further supported by trypan blue (Figure 3B), NBT (Figure 3C), and DAB (Figure 3D) staining, which indicated a higher level of cell death, more superoxide anions (O_2_¯), and H_2_O_2_ accumulation, respectively, in transgenic lines. This histochemical staining result further suggested that transgenic lines resisted *G. cichoracearum* infection by causing more cell death, which is a limiting factor to biotrophic pathogens. Our results are in agreement with Dickman and Fluhr [5], who reported that PCD followed the hypersensitive response (HR) after the pathogen invasion of a host plant. HR is induced by the host plant as the first line of defense. Also, in the case of prolonged stress conditions, there is increased reactive oxygen species (ROS) accumulation and decreased photosynthesis, which can result in the induction of downstream PCD pathways [77]. We observed that *AtICS1* and *AtPR1*, the SA-dependent disease resistance genes, were significantly upregulated in transgenic plants after PM infection (Figure 4A). This indicates that *JAZ7* may contribute to resistance to powdery mildew by promoting the expression of SA-mediated genes. Wildermuth et al. [78] reported that after pathogen invasion in Arabidopsis, the majority of the SA is produced from chorismate through the isochorismate pathway by the expression of ICS1 and ICS2. Secondly, post-pathogen invasion is a hypersensitive response that increases SA levels to trigger SAR. On the other hand, *AtLOX3* and *AtPDF1.2* were expressed to lower levels after PM infection (Figure 4B), suggesting the suppression of JA signaling.

In contrast, in the case of *B. cinerea*, fewer disease symptoms were observed on nontransgenic control plants than on transgenic lines expressing *VqJAZ7* at 72 hpi (Figure 5A). We also found significantly larger lesions on transgenic lines as compared to WT plants (Figure 5D). Previous studies have demonstrated that necrotrophic fungi induced PCD (e.g., *Cochliobolus victoriae*, a causal agent of Victoria blight disease in oats) [79]. After an attack by a pathogen, plants induce HR as the first line of defense, but this HR is utilized by the necrotrophs for their benefit, because HR has been proved to increase pathogenicity by offering a growth substrate to the *B. cinerea* [80]. We also measured the transcript levels of defense-related genes, including *AtPR1*, *AtICS1*, *AtPDF1.2*, and *AtLOX3* in response to *B. cinerea* at 0 hpi, 24 hpi, 48 hpi, and 72 hpi, (Figure 6), to gain understanding of the pathways by which resistance may be mediated. After inoculation with *B. cinerea*, *AtICS1* was expressed to ~9.9-fold and sevenfold higher in *VqJAZ7* transgenic lines compared to control plants at 48 hpi and 72 hpi, respectively. Likewise, the expression of *AtPR1* was elevated 4.4-fold and 4.1-fold at 48 hpi and 72 hpi, respectively, in the transgenic plants (Figure 6). In contrast, *AtPDF1.2* and *AtLOX3* were expressed to lower levels in transgenic lines as compared to controls (Figure 6) at all of the time points post-infection. *AtPDF1.2* and *AtLOX3* were both expressed differentially, but this expression was lower than in nontransgenic control plants. Several studies showed that in the case of necrotrophic pathogen attack, plants defend themselves by deploying JA-dependent defense responses [81,82], but on the other hand, several studies have shown that SA antagonizes JA-dependent defenses when accumulated endogenously, and subsequently prioritize SA-dependent resistance to JA-dependent defense [83]. PDF1.2 is strongly suppressed by SA [84,85], and this antagonism was also observed in a large number of accessions in Arabidopsis [86]. Contrasting roles were noted for JA signaling and JA-mediated defense in Arabidopsis to *F. oxysporum* resistance, e.g., the mutation of MYC2 and LBD20, which are transcription factors, or the overexpression of the *ERF1* and *AtERF2*, JA-defense activators, resulted in the upregulation of JA-dependent defense genes and increased resistance to *F. oxysporum* [87,88]. On the other hand, non-defensive aspects of JA-mediated senescence enhanced susceptibility to this pathogen [88,89]. Thatcher et al. [89] studied this closely, and proposed that both defensive and non-defensive JA-signaling are activated following *F. oxysporum* infection in wild-type plants, but with a significant role of non-defensive aspects to disease outcome.

Active PCD promotes the infection of *B. cinerea* in the host [90]. In the case of a necrotrophic pathogen such as *B. cinerea*, ROS exhibited susceptibility in the host plant, and its accumulation at later time points directly benefited the fungus [91]. From previous studies, it is evident that *B. cinerea* generates ROS in plants as a mechanism to facilitate its invasion [90,92,93]. We also noted more cell death (Figure 5B) and higher levels of H_2_O_2_ (Figure 5C) in transgenic lines compared with control plants, indicating that *B. cinerea* induced cell death and produced more H_2_O_2_. We also noted that the activities of antioxidant enzymes were lower in the *VqJAZ7* transgenic lines than in the nontransgenic control lines, which further indicated that the heterologous expression of *VqJAZ7* promoted *B. cinerea* disease progression.

*VqJAZ7* transgenic plants were also less resistant to infection by *Pst* DC3000 (Figure 7A). Histochemical staining with trypan blue (Figure 7B) and NBT (O_2_^−^) (Figure 7C) indicated high levels of cell death and the accumulation of more superoxide anions in nontransgenic control plants as compared to *VqJAZ7* transgenic lines when infected with *Pst* DC3000. Some pathogens developed the capacity to produce hormone mimics [87], which imitate the phytohormones in structure and function to hijack the defense regulatory network of plants for their own benefit. This hormonal divergence activates improper defense responses. For example, *P. syringae* produces coronatine, which is a mimic of JA-Ile, hence suppressing SA-mediated defense responses and promoting disease symptoms in Arabidopsis [22,89]. It is evident from genetic studies that jasmonates antagonize SA-mediated plant defenses [94,95]. In the same way, in our study, both SA-dependent and JA-dependent genes were downregulated after *P. syringae* infection. Coronatine production by *P. syringae* might not only suppress SA-mediated gene expression, but also repress the expression of JA-mediated genes due to the structural and functional similarity between coronatine and *JA-Ile*. Several studies have suggested that coronatine can suppress local and systemic defenses, resulting in tissue chlorosis and necrosis [96,97,98,99]. We also challenged the *VqJAZ7* transgenic plants with defense-response elicitors, including the peptide flg22 [100], and lipopolysaccharide, which serves as a stimulant of innate immunity [101]. We found that the transgenic plants accumulated less callose than controls after the injection of *Pst* DC3000, LPS and flg22 (Figure 9), which further validates that the overexpression *VqJAZ7* resulted in susceptibility to this pathogen.

Finally, our result concludes that *VqJAZ7* from the disease-resistant, Chinese wild grapevine, *V. quinquangularis* cv. “Shang-24”, can increase resistance to plant diseases. We found that transgenic Arabidopsis lines expressing *VqJAZ7* enhanced resistance to *G. cichoracearum* and activated SA signaling. We also found enhanced susceptibility against *B. cinerea* and *Pst* DC3000, which may be because of the *VqJAZ7* transgenic lines’ antagonistic effect on SA-dependent and JA-dependent pathways. Our study shows that *VqJAZ7* is activated by biotic stress. Future research will investigate the transcriptional networks and underlying regulatory mechanisms associated with *VqJAZ7* defense responses to different pathogens.

## Figures and Tables

**Figure 1 ijms-19-03889-f001:**
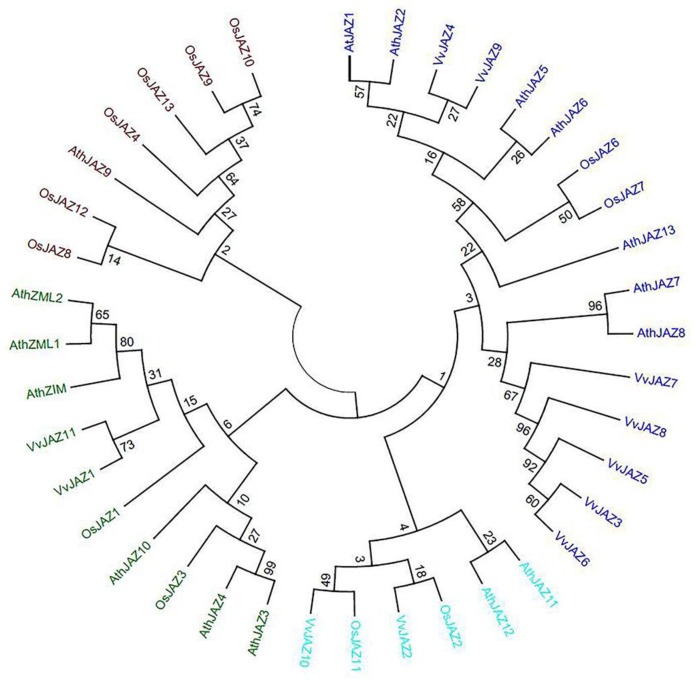
Phylogenetic analysis of Jasmonate ZIM-domain (JAZ) proteins from grapevine, Arabidopsis, and rice. Sequences were aligned and the phylogenetic tree was generated using the neighbor-joining method in ClustalX2 (http://www.clustal.org/clustal2/). The tree was designed using the online tool Interactive Tree of Life (Version 3 (http://itol.embl.de/). The bootstraps values were calculated with 1000 replicates.

**Figure 2 ijms-19-03889-f002:**
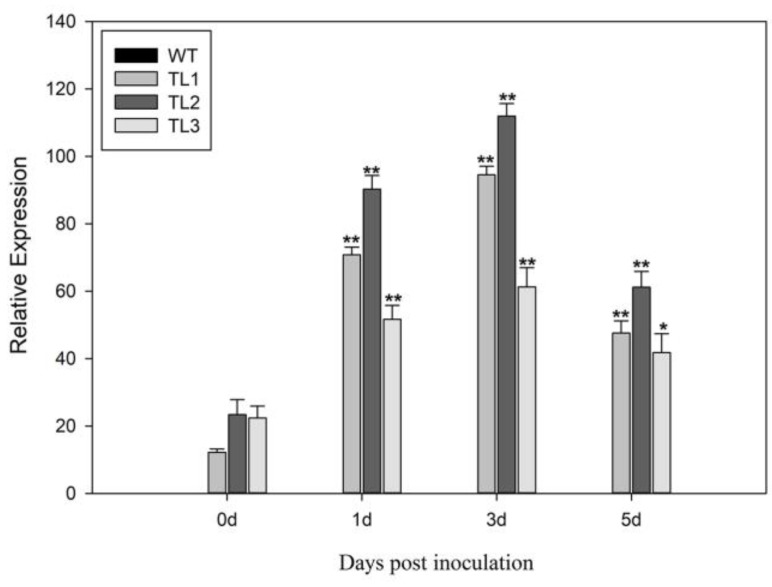
Expression of *VqJAZ7* following inoculation with *G. cichoracearum.* The *VqJAZ7* expression levels post *G. cichoracearum* inoculation in nontransgenic Arabidopsis and transgenic lines TL1, TL2, and TL3. The *VqJAZ7* expression levels increased gradually from 0 days post-inoculation (dpi) and peaked at three dpi. The *VqJAZ7* transcript levels were 0 in wild-type (WT) plants. The data are means ± SD from three experiments. Asterisks indicate statistical significance (**: *p* < 0.01, *: *p* < 0.05, one-way analysis of variance (ANOVA)).

**Figure 3 ijms-19-03889-f003:**
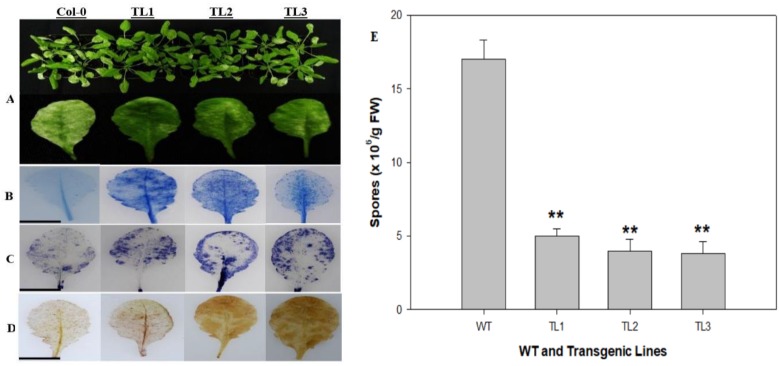
Effects of heterologous *VqJAZ7* expression in transgenic Arabidopsis on powdery mildew symptoms. *VqJAZ7* transgenic lines (TL1, TL2, and TL3) and nontransgenic control (Col-0) were infected with powdery mildew, and disease symptoms were visualized at seven dpi. (**A**) Images of aerial parts (above) and representative leaves (below) of plants at seven dpi; (**B**–**D**) Trypan blue, nitro blue tetrazolium (NBT), and diaminobenzidine (DAB) staining were carried out on leaves five dpi in order to detect cell death (**B**); superoxide anion accumulation (**C**); and H_2_O_2_ (**D**); respectively. Scale bars = 10 mm; (**E**) Spore number per gram of fresh leaves. Data bars are means ± SD from three independent experiments. Asterisks show statistical significance between overexpressing lines and nontransgenic control (** *p* < 0.01, Student’s *t*-test).

**Figure 4 ijms-19-03889-f004:**
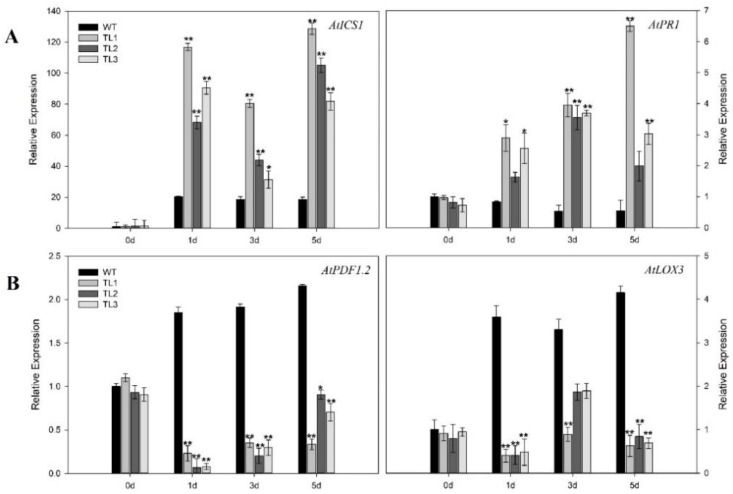
Relative expression of defense-related genes SA-dependent (**A**) and JA-dependent (**B**) in PM-infected leaves at zero, one, three, and five days post-inoculation (dpi) via qPCR. Data represent mean values ± SD from three independent experiments. Asterisks indicate significant differences between nontransgenic control plants (Col-0) and transgenic lines (TL) at different time points (**: *p* < 0.01, *: *p* < 0.05, Student’s *t*-test).

**Figure 5 ijms-19-03889-f005:**
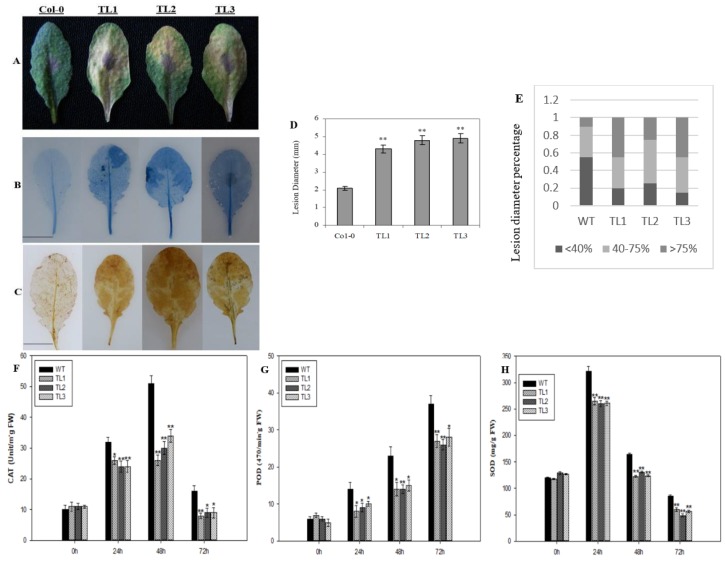
Effects of heterologous *VqJAZ7* expression in transgenic Arabidopsis on *B. cinerea* symptoms and defense-related enzyme activity. *VqJAZ7* transgenic lines (TL1, TL2, and TL3) and nontransgenic controls (Col-0) were infected with *B. cinerea* (**A**) Images of plants at 72 h post-inoculation (hpi). Trypan blue and DAB staining were carried out on leaves 72 hpi in order to detect cell death (**B**); and H_2_O_2_ (**C**), respectively. Scale bars = 10 mm; (**D**) *Botrytis* average lesion diameter at 72 hpi; (**E**) Symptoms on transgenic and control lines 72 hpi with *B. cinerea*. Activity of catalase (CAT) (**F**); peroxidase (POD) (**G**) and superoxide dismutase (SOD) (**H**) in the leaves of transgenic and WT plants. Asterisks indicate significant difference between nontransgenic (NT) and transgenic lines (**: *p* < 0.01, *: *p* < 0.05, Student’s *t*-test).

**Figure 6 ijms-19-03889-f006:**
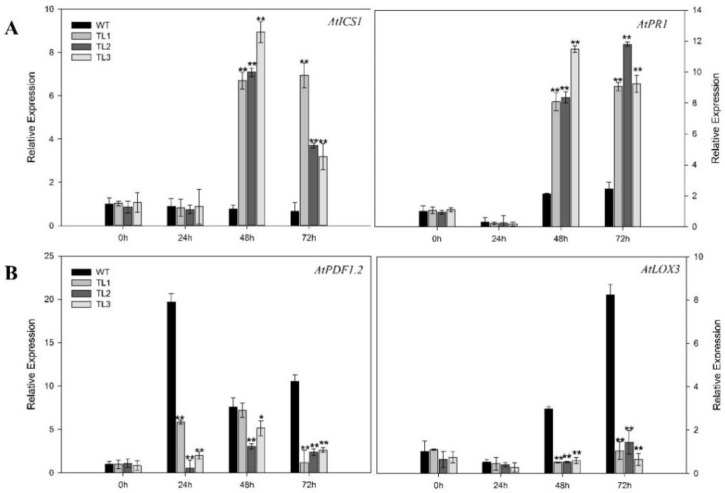
Relative expression of defense-related genes in SA-dependent (**A**) and JA-dependent (**B**) leaves infected by *B. cinerea* at 0 hpi, 24 hpi, 48 hpi, and 72 hpi. Expression was evaluated using qPCR. Data represent mean values ± SD from three independent experiments. Asterisks indicate significant differences between transgenic lines and controls (**: *p* < 0.01, *: *p* < 0.05, Student’s *t*-test).

**Figure 7 ijms-19-03889-f007:**
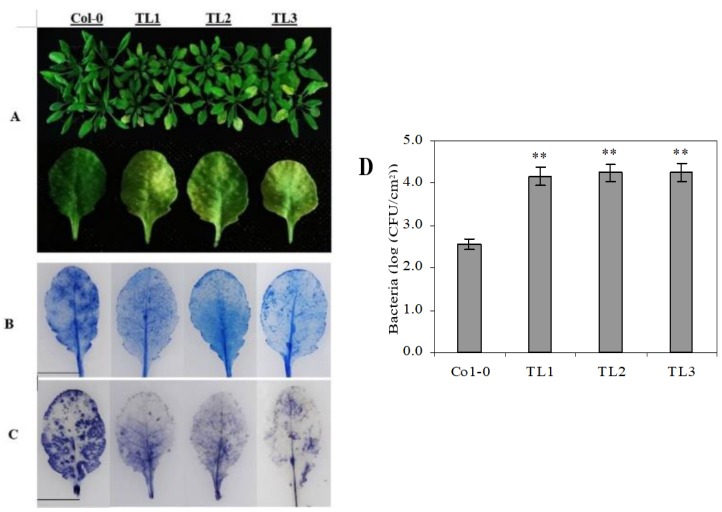
Effects of heterologous *VqJAZ7* expression in transgenic Arabidopsis on *PST* DC3000 pathogenicity. *VqJAZ7* transgenic lines (TL1, TL2, and TL3) and nontransgenic controls (Col-0) were infected with DC3000. (**A**) Images of aerial portions of whole plants (above) and representative leaves (below) at 72 hpi; (**B**, **C**) Assays for cell death (**B**) and superoxide (**C**) at 72 hpi using trypan blue and NBT staining, respectively. Scale bars = 10 mm; (**D**) *Pst* DC3000 bacterial population at 72 hpi. Asterisks indicate significant difference between control and transgenic lines (** *p* < 0.01, Student’s *t*-test).

**Figure 8 ijms-19-03889-f008:**
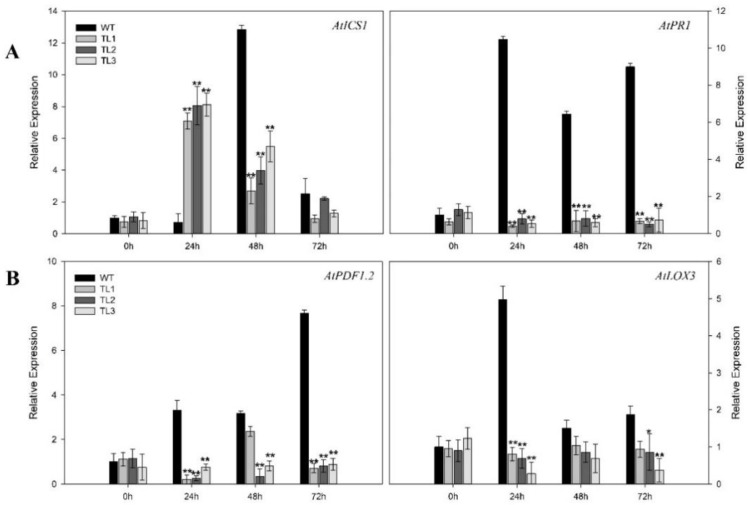
Relative expression of defense-related genes SA-dependent (**A**) and JA-dependent (**B**) in leaves infected by *Pst* DC3000 at 0 hpi, 24 hpi, 48 hpi, and 72 hpi. Expression was evaluated using Q-RTPCR. Data represent mean values ± SD from three independent experiments. Asterisks indicate significant differences between control and transgenic lines (**: *p* < 0.01, *: *p* < 0.05, Student’s *t*-test).

**Figure 9 ijms-19-03889-f009:**
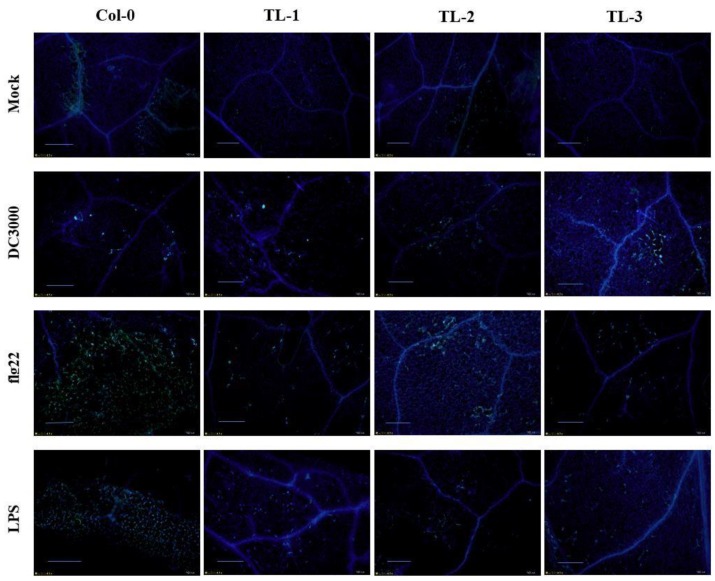
Effects of heterologous *VqJAZ7* expression in transgenic Arabidopsis on callose deposition. All samples were stained with aniline blue after *Pst* DC3000, flg22, or LPS application. All of the experiments were repeated three times with six infected leaves per repetition. The scale bar indicates 100 µm.

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
