# Peer review of "Heterologous Expression of the Grapevine JAZ7 Gene in Arabidopsis Confers Enhanced Resistance to Powdery Mildew but Not to Botrytis cinerea"

_ijms, 2018, doi:10.3390/ijms19123889_

Round 1

Reviewer 1 Report

Dear Authors,

Reviewer comments ijms-387739

Reviewer comments ijms-387739

The manuscript entitled „Heterologous expression of the grapevine JAZ7 gene in Arabidopsis confers enhanced resistance to powdery mildew but not to Botrytis cinerea“ represents a useful study aimed at an effect of heterologous expression of JAZ7 gene from grapevine in Arabidopsis plants with respect to powdery mildew or Botrytis cinerea infections. The experimental data are appropriately documented showing both Arabidopsis wild-type and three transgenic lines expressing VqJAZ7 gene under CaMV35S promoter which were exposed to several days or hours to a given pathogen inoculum (0 days = control variant with no pathogen). The presented data have appropriate statistical evaluation and appropriate interpretation in Discussion section. I have only a few minor comments on the manuscript.

1/ In Materials and methods, the source of all plant and pathogen biological materials used (Vitis quiquangularis cv. Shang-24; Arabidopsis thaliana Col-0 plant; inoculum of Golovinomyces cichoracearum, Botrytis cinerea, Pseudomonas syringae) has to be specified in the section 3.1. Plant and pathogen material and growth conditions.

2/ In Figure 1 legend to a phylogenetic tree of JAZ proteins from Arabidopsis, grapevine and rice, the authors write that „The bootstrap values were calculated with 1000 replicates.“; however, no bootstrap values are shown in Figure 1. The authors have to add visible numbers corresponding to bootstrap values to each node in the phylogenetic tree in Figure 1.

3/ In Figure 5A, B, C, a scale bar has to be added to the photographs of infected Arabidopsis leaves analogously to Figure 3 and Figure 7.

4/ In Figure 9, the fluorescent images of callose are barely visible for the reader. The authors should try to improve the quality of the images and to provide more contrast between the callose and the background in the microphotographs.

Otherwise, I have only a few formal comments on the manuscript.

 5/ Abbreviations „dpi“, i.e., „days post inoculation“, and „hpi“, i.e., „hours post inoculation“, have to be explained when first used in the manuscript text.

6/ Line 92: The section „2. Results and Discussion“ has to be renamed as „2. Results“ only since there is a separate Discussion section in the manuscript.

7/ When a reference is the subject of a sentence, the first author´s name has to be given in the reference, i.e., Discussion, line 369: „Wildermuth et al. [78] reported that,…“ (NOT: „[78] reported that,…“).

8/ Minor formal comments related to English language:

Results, line 94: Replace the word „on“ by the word „in“ in the sentence „To gain insight in potential functional relationships among JAZ proteins,…“

Results, line 190; Discussion, line 360, and further in the manuscript text: Add a comma both preceding and following (before and after) the word „respectively.“, i.e., „Likewise, expression of AtPR1 was elevated 4.4-fold and 4.1-fold at 48 and 72 hpi, respectively, in the transgenic plants.“

Line 360: „…more superoxide anions and hydrogen peroxide accumulation, respectively,…“

Final recommendation: Accept after a minor revision.

Author Response

Reviewer 1:

1) In Materials and methods, the source of all plant and pathogen biological materials used (Vitis quiquangularis cv. Shang-24; Arabidopsis thaliana Col-0 plant; inoculum of Golovinomyces cichoracearum, Botrytis cinerea, Pseudomonas syringae) has to be specified in the section 3.1. Plant and pathogen material and growth conditions.

Response: Thank you very much for your valuable suggestions. We agree to your comments and we specified the sources information in section 3.1 in our revised manuscript.

2) In Figure 1 legend to a phylogenetic tree of JAZ proteins from Arabidopsis, grapevine and rice, the authors write that „The bootstrap values were calculated with 1000 replicates.“; however, no bootstrap values are shown in Figure 1. The authors have to add visible numbers corresponding to bootstrap values to each node in the phylogenetic tree in Figure 1.

Response: We agree to your comment and we replaced the Figure to show the bootstrap values to each node.

3) In Figure 5A, B, C, a scale bar has to be added to the photographs of infected Arabidopsis leaves analogously to Figure 3 and Figure 7.

Response: According to your comment we added the scale bars to Figure 5A, B and C.

4) In Figure 9, the fluorescent images of callose are barely visible for the reader. The authors should try to improve the quality of the images and to provide more contrast between the callose and the background in the microphotographs.

Response: Thank you for your valuable comment. We agree to your comment and we replaced some images with improved contrast.

Otherwise, I have only a few formal comments on the manuscript.

 5) Abbreviations „dpi“, i.e., „days post inoculation“, and „hpi“, i.e., „hours post inoculation“, have to be explained when first used in the manuscript text.

Response: We explained the abbreviations when first used in the revised manuscript (Lines: 114 and 168).

6) Line 92: The section „2. Results and Discussion“ has to be renamed as „2. Results“ only since there is a separate Discussion section in the manuscript.

Response: The section “Results and Discussion” has been renamed as “Results” in our revised manuscript.

7) When a reference is the subject of a sentence, the first author´s name has to be given in the reference, i.e., Discussion, line 369: „Wildermuth et al. [78] reported that,…“ (NOT: „[78] reported that,…“).

Response: We rewrite the sentences according to your suggestion in line 374 and line 404 in our revised manuscript.

8) Minor formal comments related to English language:

Results, line 94: Replace the word „on“ by the word „in“ in the sentence „To gain insight in potential functional relationships among JAZ proteins,…“

Response: We replaced the word “on” by the word “in” in line 95 in our revised manuscript.

Results, line 190; Discussion, line 360, and further in the manuscript text: Add a comma both preceding and following (before and after) the word „respectively.“, i.e., „Likewise, expression of AtPR1 was elevated 4.4-fold and 4.1-fold at 48 and 72 hpi, respectively, in the transgenic plants.“

Line 360: „…more superoxide anions and hydrogen peroxide accumulation, respectively,…“

Response: We added a comma both preceding and following the word “respectively” in the revised manuscript according to your suggestion in lines no. 155, 176, 192, 193, 279, 364, 390, and 391.

Reviewer 2 Report

This manuscript reports that overexpression of grapevine JAZ7 gene (VqJAZ7) in Arabidopsis altered resistance to pathogens. VqJAZ7 expressing Arabidopsis shows enhanced resistance to Golovinomyces cichoracearum, which is biotrophic fungus. On the other hand, VqJAZ7 expressing Arabidopsis shows suppressed resistance to Botrytis cinerea, which is necrotrophic fungus. Results of these resistance tests are very clear, but these results are expected ones in my opinion. Plants usually protect themselves from biotrophic pathogen using SA pathway, which are antagonistic to JA pathway. Because JAZ is repressor of JA signaling, JAZ overexpressing plants are expected to show increased signal of SA and enhanced resistance to biotrophic pathogen. Similarly, JAZ overexpressing plants are expected to show decreased signal of JA and suppressed resistance to necrotrophic pathogen. VqJAZ7 expressing Arabidopsis also shows the suppressed resistance to Pseudomonas syringae with suppression of both SA and JA pathway. This result is surprising and interesting one. However, the mechanism underlying this result is not well investigated or discussed. From these results, I cannot recommend the publication of the manuscript in present form. I listed the comments should be addressed before publication.

MAJOR COMMENTS

1. Why both SA and JA pathway are suppressed in VqJAZ7 expressing Arabidopsis upon P. syringae infection? The manuscript describes that coronatine might repress JA pathway (line 423). Does coronatine act as an agonist of JA-Ile? Are there some reports that express antagonistic function of coronatine against JA-Ile?

Is not altered coronatine-induced expression of JA-responsive genes in VqJAZ7 expressing Arabidopsis?

2. The manuscript should show that interaction of VqJAZ7 with AtCOI1 and AtMYC2. This investigation will give the clue to understanding the suppression mechanism of JA- and SA-signaling.

MINOR COMMENTS

1. The result of gel electrophoresis of VqJAZ7 vector should be omitted (Fig. 2a). It is just preparation of materials.

2. In line 432-434, the fact of enhanced resistance against G. cichoracearum is described. However, VqJAZ7 expressing Arabidopsis shows suppressed resistance to Botrytis cinereal and P. syringae. These facts are also important. I feel that the manuscript is not touching inconvenient facts.

The enhancement of SA-signaling is trade-off to suppression of JA-signaling. Thus, these contrasting results are reasonable.

Author Response

Reviewer 2:

1) Why both SA and JA pathway are suppressed in VqJAZ7 expressing Arabidopsis upon P. syringae infection? The manuscript describes that coronatine might repress JA pathway (line 423). Does coronatine act as an agonist of JA-Ile? Are there some reports that express antagonistic function of coronatine against JA-Ile? Is not altered coronatine-induced expression of JA-responsive genes in VqJAZ7 expressing Arabidopsis?

Response: Thank you very much for your valuable suggestions. We cite some references to answer your questions in the following lines. We apologize that we could not refer these earlier. Microarray analysis reveals that of 41 JA-response genes, three are involved in signaling pathways for ethylene, auxin, and salicylic acid, confirming the interaction between JA signaling and other signaling pathways (Sasaki et al., 2001). In our study, the coronatine might induce SA biosynthesis as the expression of AtICS1 was upregulated. But this expression was suddenly dropped on the subsequent time points (48 and 96 hpi) and did not resist the pathogen. Spoel et al. 2007 proposed that tradeoff between biotroph and necrotroph resistance requires a certain threshold level of SA relative to JA. They used the Pst DC3000 cmaA/cfa6 double mutant and found that coronatine deficiency in this double mutant did not affect cross-talk in systemic tissue at the level of gene expression or disease resistance.

Therefore, it is more plausible that tradeoff requires high concentrations of SA. Indeed, it was recently reported that high SA concentrations antagonized JA-induced gene expression, whereas low levels of SA were less effective in this respect (Mur et al. 2006).

2) The manuscript should show that interaction of VqJAZ7 with AtCOI1 and AtMYC2. This investigation will give the clue to understanding the suppression mechanism of JA- and SA-signaling.

Response: Thank you very much for your valuable comment. We cite some references to answer your questions in the following lines. We apologize that we could not refer these earlier.

Using Y2H approaches, Thatcher et al. 2016 used      JAZ5 and JAZ8 as positive controls as both interact with MYC2, MYC3 and      MYC4 in all published studies (Cheng et al., 2011; Fernandez-Calvo et al.,      2011) and they found a strong interaction between JAZ7-MYC3 and JAZ7-MYC4,      but failed to identify a JAZ7-MYC2 interaction.

To our knowledge, according to the literature, the COI1-binding      capacity of the JAZ7 Jas motif is unknown. The N-terminal domain of the      Jas motif was identified as the COI1 and JA-Ile/COR binding site and      termed the JAZ degron. A LPIARR sequence in the JAZ degron binds JA-Ile      and COI1 in a clamp (Sheard et al., 2010). This sequence is diverged in      JAZ5, JAZ6, JAZ7 and JAZ8, with both JAZ7 and JAZ8 lacking the RR or RK      amino acid combination to be critical for COI1 binding (Melotto et al.,      2008; Sheard et al.,      2010; Shyu et al., 2012). Shyu et al. (2012) attributed the JAZ8 Jas      motif’s divergence from the canonical JAZ degron to its very weak ability      to associate with COI1. Because of high similarity to the divergent JAZ8      Jas motif (Shyu et al. 2012), it is likely that JAZ7 either does      not or only weakly associates with COI1.

Indeed, JAZ7 acts as repressor and can bind to both transcriptional      activators (e.g. MYC3) and repressors (e.g. JAM1) of JA-responses      (Thatcher et al. 2016), suggest that JA-sensitivity in jaz7-1D may      result from the high and/or ectopic levels of JAZ7 inhibiting normal      COI1-JAZ-TPL-TF interactions and highlights one of the difficulties in      dissecting the individual roles of proteins that act within multiprotein      complexes. In summary, the unusual protein binding properties of JAZ7 compared to other JAZs (for      example, lack of homo- or heterodimerization and divergent JAZ degron) may      lead to the JA/COR suppression.

Moreover, in our lab, we started research on JAZ gene family regarding different aspects of biotic and abiotic stresses. Our work is in the initial stage and the suggested investigation will definitely be a part of our next experiment which is in progress, please.

MINOR COMMENTS

1) The result of gel electrophoresis of VqJAZ7 vector should be omitted (Fig. 2a). It is just preparation of materials.

Response: We agree to your suggestion hence omit the Figure 2a.

 2) In line 432-434, the fact of enhanced resistance against G. cichoracearum is described. However, VqJAZ7 expressing Arabidopsis shows suppressed resistance to Botrytis cinereal and P. syringae. These facts are also important. I feel that the manuscript is not touching inconvenient facts.

The enhancement of SA-signaling is trade-off to suppression of JA-signaling. Thus, these contrasting results are reasonable.

Response: We agree to your comment and add a description for B. cinerea and P. syringae

Thank you once again for your comments and suggestions.

References:

1. Cheng, Z.; Sun, L.; Qi, T. et al. The bHLH transcription factor MYC3    interacts with the jasmonate ZIM-domain proteins to mediate jasmonate response in arabidopsis. Mol. Plant 2011, 4, 279–288.

2. Fernández-Calvo, P.; Chini, A.; Fernández-Barbero, G. et al. The Arabidopsis      bHLH Transcription Factors MYC3 and MYC4 Are Targets of JAZ Repressors and Act Additively with MYC2 in the Activation of Jasmonate Responses. Plant Cell 2011, 23, 701–715.

3.Melotto, M.; Mecey, C.; Niu, Y. et al. A critical role of two positively charged amino acids in the Jas motif of Arabidopsis JAZ proteins in mediating coronatine- and jasmonoyl isoleucine-dependent      interactions with the COI1 F-box protein. Plant J. 2008, 55, 979–988.

4.Sheard, L. B.; Tan, X.; Mao, H.; et al. Jasmonate perception by inositol phosphate-potentiated COI1- JAZ co-receptor. Nature 2011, 468, 400–405.    

5.Spoel, S. H.; Johnson, J. S.; Dong, X. Regulation of tradeoffs between plant defenses against      pathogens with different lifestyles. Proc. Natl. Acad. Sci. 2007, 104, 18842–18847.

6. Thatcher, L. F.; Cevik, V.; Grant, M.; et al. Characterization of a JAZ7 activation-tagged Arabidopsis mutant with increased susceptibility to the fungal pathogen Fusarium oxysporum. J. Exp. Bot. 2016, 67, 2367–2386.

Round 2

Reviewer 2 Report

The revised manuscript has been addressed the issues raised in previous review. Now, the manuscript is suitable for publication.